# A Chromosome-Length Assembly of the Hawaiian Monk Seal (*Neomonachus schauinslandi*): A History of “Genetic Purging” and Genomic Stability

**DOI:** 10.3390/genes13071270

**Published:** 2022-07-18

**Authors:** David W. Mohr, Stephen J. Gaughran, Justin Paschall, Ahmed Naguib, Andy Wing Chun Pang, Olga Dudchenko, Erez Lieberman Aiden, Deanna M. Church, Alan F. Scott

**Affiliations:** 1Department of Genetic Medicine, Johns Hopkins University School of Medicine, Baltimore, MD 21287, USA; dwmohr@jhmi.edu (D.W.M.); jpascha5@jhmi.edu (J.P.); 2Department of Ecology & Evolutionary Biology, Princeton University, Princeton, NJ 08544, USA; stephengaughran@gmail.com; 3Bionano Genomics, Inc., 9640 Towne Centre Dr., Suite 100, San Diego, CA 92121, USA; ahmed.naguib@gmail.com (A.N.); apang@bionanogenomics.com (A.W.C.P.); 4The Center for Genome Architecture, Department of Molecular and Human Genetics, Baylor College of Medicine, Houston, TX 77030, USA; olga.dudchenko@bcm.edu (O.D.); erez@erez.com (E.L.A.); 5Center for Theoretical Biological Physics, Rice University, Houston, TX 77030, USA; 6UWA School of Agriculture and Environment, The University of Western Australia, Crawley, WA 6009, Australia; 7Broad Institute of MIT and Harvard, Cambridge, MA 02139, USA; 8Shanghai Institute for Advanced Immunochemical Studies, ShanghaiTech University, Shanghai 201210, China; 9Inscripta, 5500 Central Ave., Boulder, CO 80301, USA; deanna.church@gmail.com

**Keywords:** Hawaiian monk seal, pinniped, endangered species, genetic purging, genome assembly, chromosome evolution

## Abstract

The Hawaiian monk seal (HMS) is the single extant species of tropical earless seals of the genus *Neomonachus.* The species survived a severe bottleneck in the late 19th century and experienced subsequent population declines until becoming the subject of a NOAA-led species recovery effort beginning in 1976 when the population was fewer than 1000 animals. Like other recovering species, the Hawaiian monk seal has been reported to have reduced genetic heterogeneity due to the bottleneck and subsequent inbreeding. Here, we report a chromosomal reference assembly for a male animal produced using a variety of methods. The final assembly consisted of 16 autosomes, an X, and portions of the Y chromosomes. We compared variants in this animal to other HMS and to a frequently sequenced human sample, confirming about 12% of the variation seen in man. To confirm that the reference animal was representative of the HMS, we compared his sequence to that of 10 other individuals and noted similarly low variation in all. Variation in the major histocompatibility (MHC) genes was nearly absent compared to the orthologous human loci. Demographic analysis predicts that Hawaiian monk seals have had a long history of small populations preceding the bottleneck, and their current low levels of heterozygosity may indicate specialization to a stable environment. When we compared our reference assembly to that of other species, we observed significant conservation of chromosomal architecture with other pinnipeds, especially other phocids. This reference should be a useful tool for future evolutionary studies as well as the long-term management of this species.

## 1. Introduction

High-quality non-human genomes, especially mammalian genomes, are needed to better constrain the limits of naturally occurring nucleotide variation, identify conserved protein and regulatory regions that may explain the distinct morphological or physiological characteristics of species, improve our understanding of evolutionary relatedness and aid efforts for species conservation and management. The Hawaiian Islands are a biodiversity hotspot where many endemic flora and fauna have been driven to extinction or are increasingly threatened as a result of human activity. Only two mammals are native to the islands: the Hawaiian Hoary bat and the Hawaiian monk seal (*Neomonachus schauinslandi;* HMS [1]). HMS are largely confined to the Northwestern Hawaiian Islands, most of which are now part of the Papahānaumokuākea Marine National Monument with a few hundred animals found in the main Hawaiian islands. The Hawaiian monk seal survived a severe bottle-neck due to overhunting in the 19th century, but its population continued to decline throughout the 20th century. Its sister species, the Caribbean monk seal (*N. tropicalis*), was last observed in 1952 and declared extinct in 2008, and the related Mediterranean monk seal (Monachus monachus) may have only between 500–1000 surviving individuals. Following the passage of the Endangered Species Act in 1973, the National Oceanic and Atmospheric Administration (NOAA) listed HMS as endangered in 1976 and published species recovery plans in 1993 and 2007 [2]. As with other endangered species, information about heterozygosity is important to determine how best to manage recovery efforts. An earlier microsatellite study of the genetic heterogeneity for the HMS suggested a significant loss of heterozygosity as a result of the bottleneck, with perhaps as few as 23 animals contributing to subsequent generations [3]. Whole genome sequences of individuals in a population, when compared to a high-quality reference genomes, provide a tool that can help better understand the consequences of bottlenecks and the effects of subsequent inbreeding. In addition, comparing sequence differences from healthy individuals of different species can be useful in distinguishing benign versus deleterious variants [4]. Chromosomal level assemblies are also necessary to establish blocks of synteny that may reflect both the evolutionary history of groups, e.g., [5], and help understand the importance of genome architecture in determining phenotypes, e.g., [6,7].

For the reasons outlined above, we chose to create a chromosomal-level assembly for the Hawaiian monk seal which improves on our earlier work [8], and which made use of a variety of technologies and analysis methods that provided insights that allowed us to compare genome architecture between pinnipeds, make inferences about the demographic history of this species, compare our reference animal to other HMS and identify potential risks for the survival of the species. In addition to the chromosomal assemblies, we were able to use linked-read sequencing to produce long haplotype blocks for the reference animal that we could contrast with a high-quality human sample for which comparable data were available. By performing both intra- and inter-species comparisons, we gained new insight into the genetics of this species.

## 2. Materials and Methods

### 2.1. Sample

A male animal (Figure 1) was chosen so that both sex chromosomes would be available for study and that assembly of the X-chromosome would not be confounded by heterozygosity. The sample was collected under the Marine Mammal Health and Stranding Response Program Permit No. 932-1905-00/MA-009526 in August 2015. Blood (~10 mL) was collected in EDTA vacutainers, shipped to Baltimore, and processed within two days of collection. The viability of the cells was assessed by trypan blue exclusion. 1 mL of whole blood was stored in LN_2_ while 9 mL (~1.85 × 10^6^ cells/mL) was used for lymphocyte separation and subsequent DNA isolation for optical genome mapping or Hi-C library preparation. As for other non-captive animals, the sample collection was opportunistic and limited which restricted our use of certain technologies. The overall sequencing strategy is shown in Appendix A and evolved as technology and analysis tools improved during the course of this study.

### 2.2. Linked-Read Sequencing

DNA was isolated using MagAttract (Qiagen, Germantown, MD, USA) and the molecular weight assayed by pulsed-field gel electrophoresis. High molecular weight (HMW) gDNA concentration was quantitated using a Qubit Fluorometer, diluted to 1.25 ng/uL in TE buffer, denatured and combined with a Chromium bead-attached primer library and emulsification oil on a Chromium Genome Chip (10X Genomics, Pleasanton, CA, USA). Library preparation was completed following the manufacturer’s protocol (Chromium Genome v1, PN-120229). Sequencing-ready libraries were quantified by qPCR (KAPA) and their sizes assayed by Bioanalyzer (Agilent Technologies, Santa Clara, CA, USA) electrophoresis. The linked-read library was sequenced (151 × 9 × 151) using two HiSeq 2500 Rapid flow cells to generate 975 M reads with a mean read length of 139 bp after trimming. The read 2 Q30 was 87.93% and the weighted mean molecule size was calculated as 92.33 kb. The mean read depth was ~61x. The sequence was processed using Supernova software [9] which demultiplexed the Chromium molecular indexes, converted the sequences to fastq files and built a graph-based assembly. The assemblies, which diverge at “megabubbles”, consisted of two “pseudohaplotypes.” The sequence data were originally analyzed using Supernova 1.0 and repeated using v1.1, which improved gap-size estimates. Subsequently, we reanalyzed the linked-read data with Supernova version 2.1.1 on 1260.32 M reads and ~78X read depth, which increased scaffold length N50 from ~22 Mb to ~62 Mb. (Appendix A). The linked-read scaffolds were used with subsequent Hi-C, nanopore and optical genome mapping analyses.

### 2.3. Nanopore Sequencing

We used Oxford Nanopore sequencing to obtain long reads for gap filling. DNA was isolated using the agarose plug method (www.bionanogenomics.com, accessed on 20 June 2019) or with the Circulomics nanobind method (www.circulomics.com, accessed on 20 June 2019). Libraries were made using either the rapid transposase (RAD-004) or ligation methods (LSK-110) and run on a GridION. A total of 4.26 M reads were obtained with molecule N50s of approximately 32 kb producing about 7.3x total read depth. PILON v1.22 was run in gap mode for filling.

### 2.4. Bionano Genomics Optical Genome Mapping

Two separate optical genome maps were created for RE74. The second map was done to take advantage of improved chemistry and instrumentation.

Sample 1: Optical genome mapping of large DNA [10] incorporates fluorescent nucleotides at sequence specific sites, visualizes the labelled molecules and aligns these to each other and to a DNA scaffold [11]. Lymphocytes were processed following the IrysPrep Kit for human blood with minor modifications. Briefly, PBMCs were spun and resuspended in Cell Suspension Buffer and embedded in 0.6% agarose (plug lysis kit, Bio-Rad Laboratories, Hercules, CA, USA). The agarose plugs were treated with Puregene Proteinase K (Qiagen, Germantown, MD, USA) in a lysis buffer (Bionano Genomics, San Diego, CA, USA) overnight at 50 °C and shipped for subsequent processing (S. Brown, KSU). High Molecular Weight (HMW) DNA was recovered by treating the plugs with Gelase (Epicenter), followed by drop dialysis to remove simple carbohydrates. HMW DNA was treated with Nt. BspQI nicking endonuclease (New England Biolabs, Ipswich, MA, USA) and fluorescent nucleotides incorporated by nick translation (IrysPrep Labeling-NLRS protocol, Bionano Genomics, San Diego, CA, USA). Labelled DNA was imaged on the Irys platform (Bionano Genomics, San Diego, CA, USA) and more than 234, 000 Mb of image data were collected with a minimum molecule length of 150 kb. Alignment was done with Solve3.2.2_08222018 software (Bionano Genomics, San Diego, CA, USA).

Sample 2: Optical genome mapping was repeated using the DLE-1 label and the Saphyr instrument (Bionano Genomics, San Diego, CA, USA). DNA from approximately 2M lymphocytes were isolated, labelled and imaged at Bionano Genomics. Alignment to the post-Hi-C assembly was performed using Bionano Solve v3.6. As with sample 1, molecules smaller than 150 kb were excluded. The second optical genome maps were aligned to the predicted assembly resulting from the Hi-C assembly. The maps were reviewed using the online Bionano Access viewer Raw data, along with hybrid assembly files, available at NCBI (SUB11144351).

### 2.5. Arima Hi-C and 3D-DNA

Approximately 2M lymphocytes were sent to Arima Genomics (San Diego, CA, USA) for Hi-C library prep. The library was sequenced locally on a NovaSeq 6000 (Illumina, San Diego, CA, USA) to a read depth of 60.8X. The Hi-C data were aligned to the linked-read Supernova scaffolds. Hi-C genome assembly was performed using the 3D-DNA pipeline (3D-DNA v180922) [12] and the output was reviewed using Juicebox Assembly Tools [13]. The Hi-C data are available on www.dnazoo.org/assemblies/Neomonachus_schauinslandi (accessed on 20 June 2019), where they were visualized using Juicebox.js, a cloud-based visualization system for Hi-C data [14]. Genome assembly before and after the Hi-C scaffolding is shown in Appendix A.

### 2.6. Final Editing

The near final assembly from the methods described above was manually edited with respect to the second optical genome map (Bionano Access v1.6.1). This mainly adjusted a small number of inverted sequence blocks within scaffolds and altered the length of N blocks introduced by Supernova. Manual inspection of the optical genome maps aligned with the sequence frequently showed that the spacing of DLE-1 sites in blocks of long Ns corresponded to repetitive sequence. The final edited version was submitted to NCBI for annotation.

### 2.7. Haplotype Analysis

The linked-read sequence allowed phasing of sequences sharing indexes into haplotype blocks. We used short-read data aligned to the final assembly to run Long Ranger 2.2.2. Loupe ver 2.1.2 (2.4) was used to visualize the pseudohaplotypes. A Loupe file for NA12878 was available at 10X Genomics and was used to compare RE74 and human orthologous regions.

### 2.8. Quality Assessments

QUAST [15] was used to generate N50 plots and comparative metrics for the assembly methods. BUSCOv5 (Benchmarking Universal Single Copy Orthologs), a tool for assessing genome completeness [16], was run on the final assembly.

### 2.9. Conserved Synteny Analysis

We compared the HMS reference to the domestic dog, cat, and other seal assemblies using minimap2 [17] and plotted these with D-GENIES [18].

### 2.10. Annotation and QC

Annotation of the version 1 and 2 scaffolds (ASM220157v1/v2) was performed at NCBI using the Eukaryotic Genome Annotation Pipeline (https://www.ncbi.nlm.nih.gov/genome/annotation_euk/process/, (accessed on 20 June 2019)). We used BUSCO ver5 to identify shared genes predicted to be present or missing in our chromosomal assembly vs. those of other species characterized by the minimap2 alignments. RNA sequences from the blood of RE74 and skin of an unrelated animal were produced and submitted to NCBI to aid in gene annotation.

### 2.11. Short Read Sequencing and Variant Calling

For the additional 9 HMS, standard Illumina short-read libraries were prepared from flipper-punch DNA, indexed and sequenced. Alignment, variant calling, and quality control were accomplished using the DRAGEN Germline v3.7.5 pipeline on the Illumina BaseSpace Sequence Hub platform [19]. FASTQ files were aligned to the RE74 reference (ASM220157v2), producing CRAM files with duplicate reads flagged. Quality control statistics for alignment and variant calling were downloaded using the Basespace CLI, and analyzed using in-house aggregation and visualization pipelines. Once adequate coverage and data quality were confirmed, joint variant calling was performed, producing a multi-sample VCF file. Filter flags were applied using the DRAGEN Germline default hard-filter quality control parameters and identified variants which do not meet the criteria of (QUAL score < 10.41 SNV or QUAL score < 7.83 for indels) or (lod_fstar > 6.3). Systematic base calling errors were corrected by default using the Base Quality Drop Off algorithm.

### 2.12. Demographic Reconstruction

We modeled the demographic history of the species using the sequentially Markovian coalescent method as implemented in MSMC2 [20,21]. We followed the workflow provided by Schiffels and Wang [21]. This included generating a mask file for the reference genome to exclude unmappable regions, and individual mask files for each genome to exclude regions of unexpectedly high or low coverage. We then ran the unphased data of four medium-to-high coverage samples through MSMC2, with a per-generation mutation rate of 7.0 × 10^−9^ [22] and a generation time of 13 years [23].

## 3. Results

### 3.1. Genome Assembly and Quality Assessment

The completed assembly consisted of 16 autosomes, the X chromosome, portions of the Y chromosome and the mitochondrial genome. A QUAST plot (Figure 2) shows the improvement in scaffold contiguity between NCBI ASM220157v1 and v2. Changes in the NCBI annotation metrics are shown in Appendix A. Improvements may reflect both changes in the contiguity of the assembly as well as evolution of the assembly pipeline. The genome report for ASM220157v2 is available at https://www.ncbi.nlm.nih.gov/assembly/GCF_002201575.2/ (accessed on 22 June 2022). Key metrics comparing the two assemblies are shown in Appendix A. Genome quality was assessed by different methods: (1) whether the chromosome lengths matched those of a karyotype for this species, (2) how well the final assembly compared to the second optical genome map, (3) how the chromosomes were aligned to other seal genomes available at NCBI and (4) how well BUSCO analysis metrics compared to that of other seals.

By both karyotype comparison and visual inspection of the optical genome maps, the assembly was judged to be of high quality and the optical genome map data has been deposited at NCBI. A small number of unplaced scaffolds (green line in Figure 2) remained. BUSCO [16], a collection of taxa-based conserved genes, was used to identify orthologs in both the chromosomal assemblies as well as the unplaced scaffolds. Although imperfect, BUSCO gene counts can differ between groups based on whether a genome is incomplete or has errors that make predictions inaccurate. True differences between taxa can also be the result of selection, allowing genes to be gained or lost over evolutionary time. BUSCO metrics for RE74 are comparable to other genomes (Table 1), suggesting both completeness of this assembly as well as that of the other species examined.

### 3.2. Phylogenic Relationships of “Missing” BUSCO Genes

We noted that there was a pattern of predicted “missing” genes based on their relatedness (Table 2) that followed their expected phylogenetic distance as calculated from timetree.org estimates of divergence time. For example, *Mirounga angusirostris*, the taxonomically closest species for which a genome was available and with a predicted divergence date of 13.5 MY, had 188 shared “missing” genes relative to RE74 vs. 128 with the domestic cat and 99 with humans. Of the genomes compared in Table 2, the largest number of “missing” BUSCO predictions occur in the seals. We interpret this to mean that many of these genes are truly absent in these related taxa and reflect a shared history or shared biology.

### 3.3. Conservation of Chromosomal Architecture Based on Genome Alignments

Arnason [24] observed that pinniped chromosomes show “pronounced karyotypic uniformity” and have either 15 or 16 autosomes [25]. Our earlier study [8] could not definitively resolve the correct number of HMS chromosomes, but the addition of Hi-C confirmed 16 autosomes. We renumbered all Hi-C autosomes according to their length and identified the X and Y by synteny. These chromosomes were then compared to those of other seals and carnivores using minimap2 and plotted with D-GENIES [18] as shown in Figure 3. Fragmentation or rearrangement of syntenic blocks in such an analysis can occur either by errors in the assemblies or true chromosomal events. Assuming correct assemblies for RE74 and the other seal genomes, the DG plots confirm conservation between species that reflects phylogenetic relatedness.

D-GENIES plots showed near perfect chromosome-to-chromosome synteny between HMS and the Northern Elephant seal (*Mirounga angustirostris*), the taxonomically closest species for which an assembly is available, as well as the Bearded seal (*Erignathus barbatus*), both of which have 16 autosome pairs as does the Hawaiian monk seal. The Gray seal (*Halichoerus grypus*), another phocid, has a fusion of HMS chromosomes 7 and 15. The harbor seal (*Phoca vitulina)* also has one fewer autosome pair due to the same fusion. In all of the phocids, the remaining autosomes are syntenic to HMS. In the more distant Otariids, the Guadalupe fur seal (*Arctocephalus townsendi)* has 17 autosome pairs, with chromosomes 8 and 10 sharing synteny with HMS 5 and chromosomes 10 and 15 sharing with HMS 6. The same is true for the Antarctic fur seal (*Arctocephalus gazella).* The least related of the pinnipeds, the walrus (*Odobenus rosmarus)* with 15 autosome pairs, is the most distinct among the seals with portions of Oros chromosome 16 and 13 sharing synteny with HMS chromosome 5, and Oros chromosome 13 and portions of chromosome 3 sharing synteny with HMS 6. Oros chromosome 3 also shares synteny with HMS 7 and portions of Oros 15 match HMS 13 and 15. Among the two other carnivores we compared, the domestic dog and cat, the cat is most similar karyotypically despite being taxonomically more distant. Cats have 18 autosome pairs and, as shown in Appendix A, HMS 1 is the equivalent of cat NC_018731.3 and parts of NC_018724.3, HMS 3 is syntenic with parts of chr C1 (NC_018730.3) and chr A1 (NC_018723.3), HMS 4 with cat chr F2 (NC_018740.3) and part of chr C1 (NC_018730.3), HMS 5 with chr B4 (NC_018729.3) and chr E3 (NC_018738.3), HMS 7 with part of chr A1 (NC_018723.3), and HMS 12 with part of chr A2 (NC_018724.3). The dog is the most different with 38 autosome pairs with at least 68 syntenic blocks that are fragmented, rearranged, or inverted relative to the monk seal (Appendix A).

### 3.4. Heterozygosity Estimates and Phasing

We created a hard-filtered multisample VCF file for the additional ten HMS aligned to the RE74 assembly using GATK as implemented on the Illumina DRAGEN platform. Using VCFtools, we compared the RE74 SNPs to autosomal SNPs from NA12878 and noted an overall reduction of heterozygosity in RE74 that was approximately one-eighth that of the human reference (350,812/3,998,486 = 12.4%). Given the smaller size of the combined seal autosomes (~2.23 Gb) vs. NA12878 (~3.09 Gb), the SNPs per Mb average was ~157 for RE74 vs. 1262 for NA12878. Considering the caveats in making such comparisons, these numbers agree with the empirical data we observed from gene-to-gene comparisons visualized in Loupe and IGV. Long Ranger (10X Genomics, v2.2.2) statistics for RE74 indicated that 96% of SNPs were phased, the phase block N50 was 557.7 kb, and the longest phase block was 7.27 Mb. We compared phase blocks between RE74 and NA12878 in the Loupe viewer. A representative example of a 1.28 Mb phase block including the KCNAB1 and PLCH1 genes is shown in Appendix A.

### 3.5. Marked Loss of MHC Heterogeneity in HMS

Immunity-related genes have been previously studied in the Hawaiian monk seal and their lack of heterogeneity documented [26]. We explored this more thoroughly by comparing RE74 to the human gold standard and to other HMS genomes using phased variants from linked-read sequencing. Figure 4 compares heterozygosity in NA12878 for HLA-DQA1 and HLA-DQB1 to the orthologous HMS MHC genes. The same observation was made for HLA DMB and DMA (Figure 5a), and HLA-DOA and HLA-DOB (Figure 5b), where we compared RE74 and 10 other seals to four well-characterized human CEPH samples, the parents of NA12878.

### 3.6. MSMC Analysis

A Multiple Sequentially Markovian Coalescence (MSMC) analysis [16] was performed for four individual HMS (Figure 6), and it illustrates a notable decline in effective population size over time, starting around 100,000 years ago. The analysis is consistent with a Hawaiian monk seal population that has had a relatively small size (Ne < 2000) for tens of thousands of years. If true, this extended period of small population size is consistent with the low levels of heterogeneity shown above.

## 4. Discussion

Comparative genomics is producing an unprecedented amount of sequence useful to explore fundamental questions about evolutionary biology and population genetics and is increasingly useful in human genetics by improving predictions of deleterious mutations (e.g., [4,27]). The recent development of methods to characterize long DNA molecules has allowed high-quality assemblies to be produced more quickly and with greater accuracy, and it has led to the characterization of a broad cross-section of living plants and animals. In this study, we applied a variety of technologies and analysis tools to characterize the genome of an endangered species that is recovering from a population bottleneck during which the effective population size may have been as few as 23 individuals [3]. We were able to show that our reference animal, “Benny”, had very low levels of heterozygosity that was confirmed by comparing variants with ten other HMS. Additionally, we estimate that he had ~12.4% percent of the heterozygosity seen in one well-characterized human for which comparable phased genotypes were available. The reduced heterozygosity is lower than in most thus far reported mammals, and it is comparable to that observed in archaic humans in whom it has been attributed to first cousin or half-sib mating [28]. The marked loss of heterozygosity was particularly notable in orthologous MHC genes where we observed virtually no variants in HMS, in contrast to genomes from CEPH family 1463 that includes NA12878 and her parents. The consequences of reduced MHC class II variability remain to be determined, but an expectation of greater susceptibility to infectious disease may be one and may, in part, explain the susceptibility of HMS to toxoplasmosis (https://www.fisheries.noaa.gov/feature-story/toll-toxoplasmosis-protozoal-disease-has-now-claimed-lives-12-monk-seals-and-left, accessed on 4 July 2022), a parasitic disease likely the consequence of the large number of feral cats in Hawaii. A secondary consequence of the markedly reduced heterozygosity in RE74 is that phase blocks were generally smaller than in NA12878 and other humans. This was likely the consequence of insufficient variation available to combine linked reads into phased scaffolds. If so, then phasing of other genomes with markedly reduced heterozygosity may be difficult without long, highly accurate reads.

Population bottlenecks can result in a high incidence of genetic disorders due to homozygosity of deleterious recessive alleles. However, as with artificial selection, purging of rare recessives can occur relatively quickly in small populations. In contrast, large populations that have experienced a rapid decline would be expected to show residual variation that was present prior to a bottleneck. In the case of Hawaiian monk seals, we propose that allelic purging for a prolonged period, as indicated by the MSMC analysis, may be the best explanation for the extremely low heterozygosity that we and others have observed and the fact that pups with obvious genetic disorders are seldom reported. While the apparent loss of deleterious recessive alleles may be advantageous, the reduction of an allelic “reserve” may also make HMS more susceptible to environmental changes (e.g., [29]). Our demographic analysis predicts HMS had historically low populations prior to human activity, which could be a consequence of the lower productivity of tropical oceans and climatic cycles, such as El Niño, that may limit resource availability prohibiting the larger populations seen for other seal species. Evolutionary theory suggests that small populations that survive over long periods of time will become increasingly specialized to their environment at the risk of losing the ability to respond to changes, and this may partially explain the high extinction rates of other species in Hawaii. Similarly low levels of heterozygosity have been reported for other species, including the vaquita, for which Robinson et al. [30] argue that the loss of deleterious alleles has reduced their risk of inbreeding depression.

In addition to determining heterozygosity within HMS, we also compared our reference to that of other pinnipeds using both chromosomal sequence alignment and by observing the similarity of genes that were shared or lost between taxa. BUSCO metrics were useful both for assessing the quality of our assembly and also for comparing it to related species, which confirmed that all the pinniped genomes we compared were of comparable quality in terms of completeness (Table 1), and to those of the better studied domestic dog, cat and human genomes. When we compared the BUSCO “missing” genes category, we noted that they were most similar between RE74 and the taxonomically closest species, the phocids (Table 2). This suggests that many of the “missing” genes are truly absent in the earless seals and not a consequence of errors or incompleteness of the genome assembly. The loss of genes between related taxa is a well-documented phenomenon. Whether the “missing” BUSCOs we observed merely reflect a shared evolutionary history or are biologically significant will require future study, but multiple instances of gene loss from adaptations to aquatic life have been documented in cetaceans and hippos [31].

Chromosome-level assemblies are valuable not only as a measure of completeness when compared to karyotypes for a species, but they can identify blocks of synteny that may be functionally important. Our D-GENIES alignments of RE74 to other seals (Figure 3) are in agreement with their relatedness both in terms of chromosome counts and overall synteny. They also agree with previous cytogenetics studies (e.g., [25]). The genome of the domestic cat shows similarity to the pinnipeds with several autosomes completely syntenic, suggesting that seals and felids may share a conserved genome architecture with early carnivorans hypothesized to have a 2n karyotype of 38 chromosomes [32,33]. In contrast, the canines that are taxonomically more closely related to seals than felines have undergone considerable chromosomal change (e.g., [34]). The consequences of the high rate of karyotypic evolution in dogs, if any, are unclear. By sequencing a male HMS, we were able to produce both X and Y scaffolds. While the X chromosome assembled well, the Y chromosome, due to long blocks of repetitive sequence, was problematic. Nevertheless, we were able to assemble portions of the Y chromosome (Table 3) corresponding to the gene-rich region seen in other species, which showed good agreement with that of *Zalophus californianus* [35] with a total of 20 loci identified in both.

In summary, by using a variety of laboratory and analytical methods, we have produced the first chromosomal-length assembly for the endangered Hawaiian monk seal. This reference contributes to further documenting of the evolutionary history of the pinnipeds and other mammals and may provide additional understanding about the importance of genome architecture and its relationship to phenotype. This reference, a legacy to “Benny”, should also provide a useful tool for continued population studies of HMS and may help reduce the consequences of inbreeding in future management. Lastly, over the course of this study we used a variety of methods in an effort to identify which were most useful (Appendix A). Sequencing technology and assembly methods have continued to mature so that large consortia such as the Vertebrate Genome Project, the DNAzoo, the Earth Biogenome Project, the Darwin Tree of Life, and others are growing in their ambition to produce high quality genomes. Future prospects for DNA sequencing technologies are bright and the ability to sequence and assemble genomes globally, perhaps even in the field, is becoming a reality.

## Figures and Tables

**Figure 1 genes-13-01270-f001:**
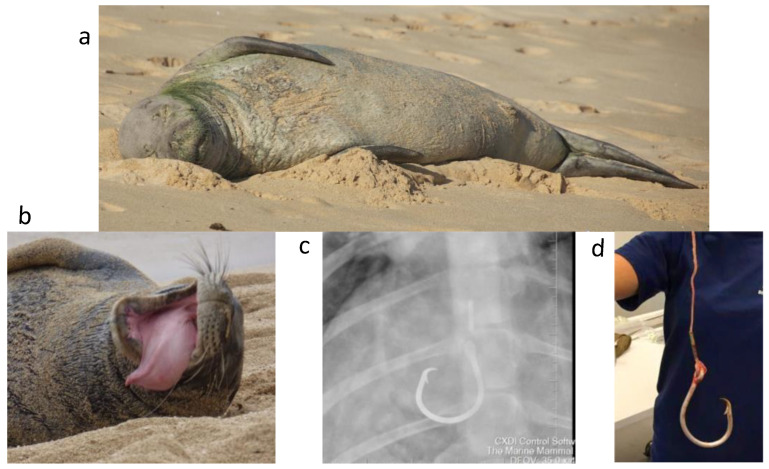
Montage of “Benny” (RE74). (**a**) Asleep on east Oahu in 2009, (**b**) In distress after swallowing a fishing hook and line (2015), (**c**) X-ray prior to surgery and (**d**) post-surgery. Images courtesy of NOAA. RE74 died of undetermined causes on 17 June 2022 at the age of 19 (https://www.fisheries.noaa.gov/pacific-islands/endangered-species-conservation/hawaiian-monk-seal-updates#saying-a-final-aloha-to-re74%C2%A0–“benny”, accessed on 22 June 2022).

**Figure 2 genes-13-01270-f002:**
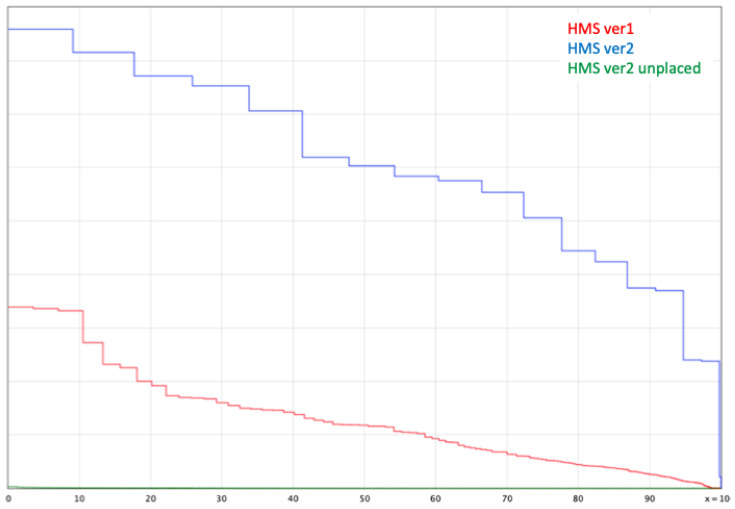
QUAST plot showing the improvement of assembly metrics from version 1 (NCBI ASM220157v1) to version 2 (NCBI ASM220157v2) and the small contribution of unplaced scaffolds in v2. The green line represents scaffolds remaining after the second optical genome mapping.

**Figure 3 genes-13-01270-f003:**
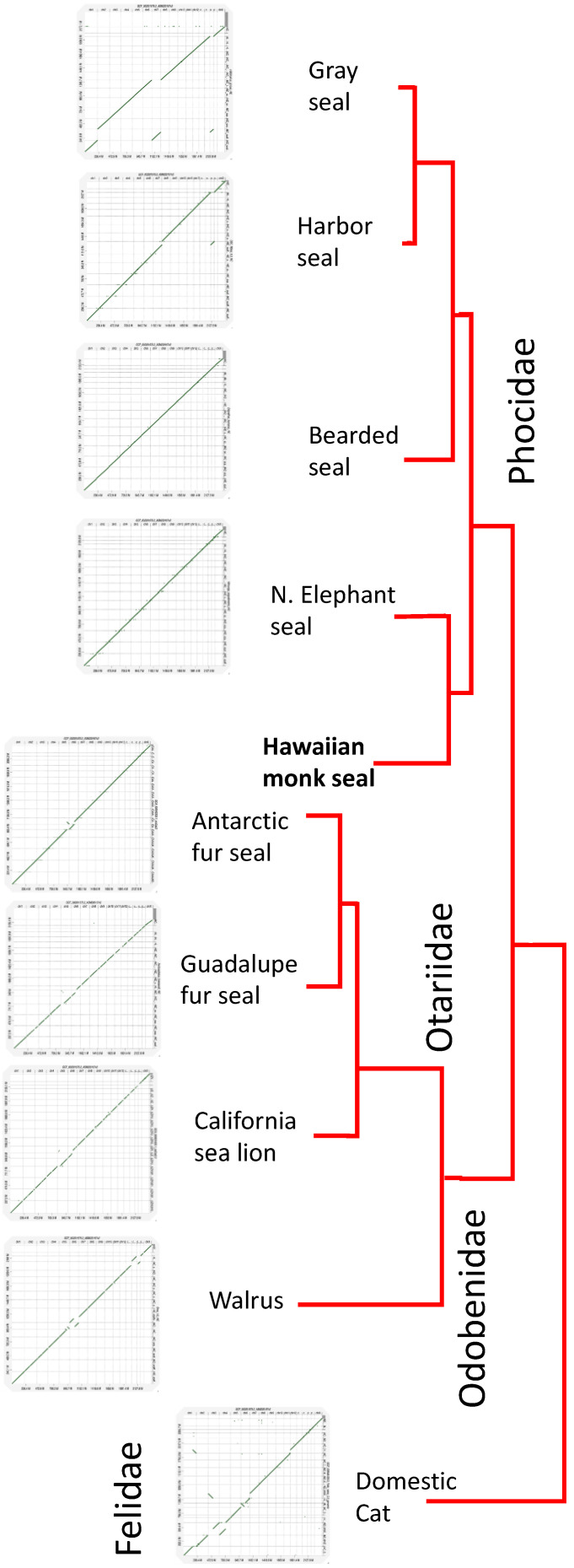
D-GENIES plots of HMS aligned to other pinnipeds for which chromosome-length assemblies are available as well as the domestic cat. The specific genomes compared are listed in Appendix A. As chromosome orientations are arbitrarily based on their order in NCBI, we reverse-complemented chromosomes in some species prior to alignment so that the chromosomes are in the same direction relative to HMS. Phocidae (earless seals), Otariidae (eared seals), Odobenidae (walrus), Felidae (felines). Enlarged individual alignments are shown in Appendix A. D-GENIES filtering was adjusted individually for each plot to maximize sequence identity but minimize small matches.

**Figure 4 genes-13-01270-f004:**
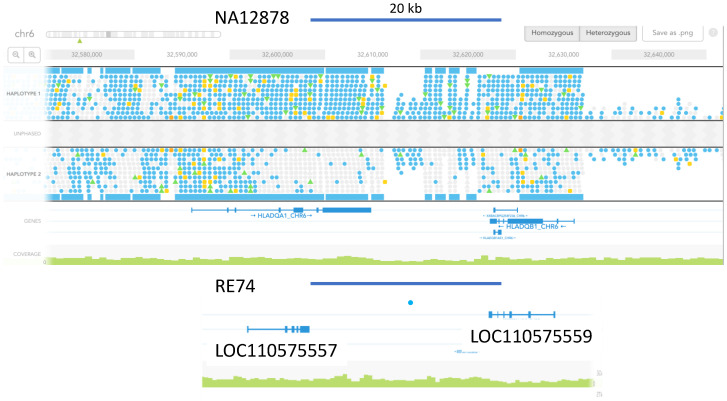
Absence of heterozygosity in RE74 at loci orthologous to human DQA1 and B1. Comparison of phased variants in NA12878 HLA DQA1 and B1 genes visualized in the Loupe viewer (10X genomics) compared to orthologous RE74 genes [26] LOC110575557 and LOC110575559. The figures were adjusted to the same scale (the blue bar represents 20 kb). Blue dots represent SNPs and yellow dots are indels. Blue squares represent multiple SNPs that are not resolved at this scale. One low quality intergenic SNV was observed in RE74.

**Figure 5 genes-13-01270-f005:**
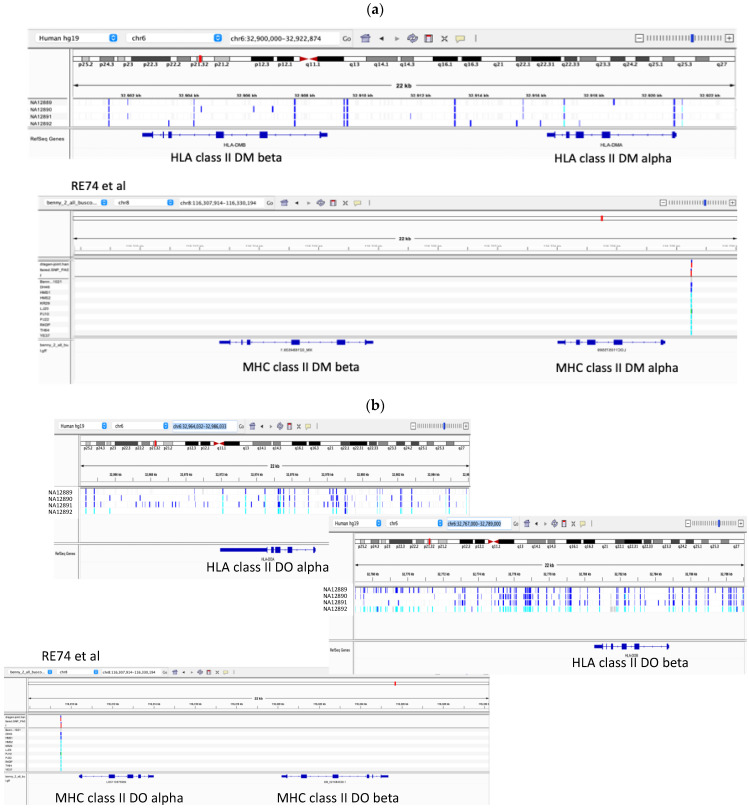
(**a**) Heterozygosity at positions flanking the MHC class II loci DMA DMB, DOA and DOB among human samples in comparison to RE74 and ten other HMS genomes as visualized in IGV. Human genomes are CEPH NA12891 and 12892, the parents of NA12878 and NA12889 and NA12890, her unrelated in-laws. A seal SNV occurs at only one position. (**b**) Comparisons of NA12889-12892 for HLA DOA and DOB (the human genes are split into two panels due to their larger intergenic sequence). Seal SNVs occur at a single position for all 10 animals for both regions. Panel width is 22 kb for all compared regions.

**Figure 6 genes-13-01270-f006:**
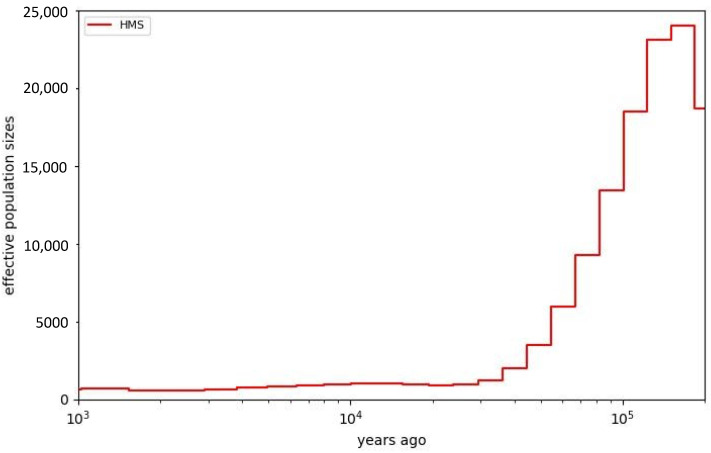
MSMC plot showing small and declining Ne over the last 200,000 years.

**Table 1 genes-13-01270-t001:** BUSCO comparisons (9226 BUSCO gene groups) of HMS to other pinniped genomes. Ma, *Mirounga angustirostris* (N. Elephant seal), Pv, *Phoca vitulina* (Harbor seal), Hg, *Halichoerus grypus* (Gray seal), Zc, *Zalophus californianus* (California sea lion), Or, *Odobenus rosmarus* (Walrus), Clf, *Canis lupus familiaris* (Domestic dog), Fc, *Felis cattus* (Domestic cat), Hs, *Homo sapiens* (GRCh38).

	RE74	Ma	Pv	Hg	Zc	Or	Clf	Fc	Hs
Complete	8845	8712	8849	8542	8878	8875	8742	8836	8856
Complete & Single-Copy	8617	8467	8651	8352	8543	8599	8560	8761	8450
Complete & Duplicated	228	245	198	190	335	276	182	75	406
Fragmented	100	175	90	242	83	82	152	111	127
Missing	281	339	287	442	265	269	332	279	243

**Table 2 genes-13-01270-t002:** Number of shared “missing” BUSCO genes between taxa. *TT est.* is the estimate divergence time of each species from HMS based on Timetree.org. Pinniped species abbreviations (bolded) are as in Table 1.

Total Missing		HMS RE74	Ma	Pv	Hg	Zc	Or	Clf	Fc	Hs
	*TT est.*	*-*	*13.5*	*18.4*	*18.4*	*26*	*26*	*46*	*54*	*96*
*281*	HMS	-	188	185	179	164	167	129	128	99
*339*	Ma		-	186	185	153	155	127	133	100
*332*	Pv			-	207	172	170	131	135	111
*332*	Hg				-	160	166	125	133	108
*265*	Zc					-	184	133	126	106
*269*	Or						-	127	134	103
*332*	Clf							-	122	104
*279*	Fc								-	106
*243*	Hs									-

**Table 3 genes-13-01270-t003:** List of Y chromosome genes identified in RE74 from NCBI annotation with homologous sequences in the *Zalophus californianus* Y (CM019820.2).

Position/Mb	Locus ID	RNA Sequence	NCBI Annotation
2.66	LOC110582119	XM_021692074.1	ubiquitin-like modifier-activating enzyme 1
2.64	LOC110582047	XM_021691967.1	BCL-6 corepressor-like
2.89	LOC110581036	XM_021690741.1	γ-taxilin-like
2.95	SRY	XM_021693078.1	sex-determining region Y, SRY
3.05	LOC110582200	XM_021692141.1	cullin-4B-like
3.33	LOC110591777	XM_021702682.1	eukaryotic translation initiation factor 1A, X-chromosomal
3.42	LOC110591779	XM_021702683.1	heat shock transcription factor, Y-linked-like
3.50	LOC110591787	XM_021702687.1	lysine-specific demethylase 5D
3.56	LOC110591776	XM_021702680.1	zinc finger X-chromosomal protein-like
3.66	LOC110591775	XM_021702679.1	eukaryotic translation initiation factor 2 subunit 3, Y-linked-like
4.41	OFD1	XM_021692108.1	OFD1, centriole and centriolar satellite protein (OFD1), partial mRNA
4.46	LOC110580574	XM_021690271.1	histone demethylase UTY-like
4.66	LOC110581163	XM_021690923.1	ATP-dependent RNA helicase DDX3X-like
4.70	LOC110581157	XM_021690911.1	probable ubiquitin carboxyl-terminal hydrolase FAF-X
4.81	LOC110580811	XM_021690464.1	RNA-binding motif protein, X chromosome-like
4.83	LOC110580801	XM_021690458.1	probable ubiquitin carboxyl-terminal hydrolase FAF-Y
4.89	LOC110580822	XM_021690474.1	serine protease 55-like
4.91	LOC110580830	XM_021690484.1	serine protease 52-like
5.17	LOC110581101	XM_021690826.1	thymosin β-4-like
5.25	LOC110584133	XM_021694172.1	ras-related protein Rab-9A-like

## Data Availability

Data related to genome assembly is available at NCBI BioProject PRJNA392091.

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
