# Peer review of "A Chromosome-Length Assembly of the Hawaiian Monk Seal (Neomonachus schauinslandi): A History of “Genetic Purging” and Genomic Stability"

_genes, 2022, doi:10.3390/genes13071270_

Round 1

Reviewer 1 Report

I have reviewed the manuscript “A chromosome-length assembly of the Hawaiian Monk seal (Neomonarchus schauinslandi): A history of “genetic purging” and genomic stability”submitted to “Genes”. This manuscript fits well within the scope of the journal; it needs some improvements; there are a few suggestions that authors may consider to improve it further:

The use of English language is reasonable, however, there are a number of punctuation and grammatical errors; that should be corrected and rephrased using academic English for a better flow of text for the reader.

- Abstract is appropriate, however the findings are not clear, some of the results can be summarized in the abstract.

The authors should define all the abbreviations at their first appearance in the text and should be used in the text.

The introduction; section is very brief and deficit, it should be expanded to further address the background information and the rationale of the study effectively.

Figure 3 and 5 images are too small, is it possible to enlarge a but further for further clarity.

Add limitations of the study if any

A section describing the conclusions should be added after the discussion.

Author Response

Thank you for your comments.  We have expanded are rewritten the Abstract, Introduction and Discussion to make the text more readable and added conclusions to the latter.  Defining the first use of abbreviations has been checked and corrected.  We agree that Fig 3 is small which is why we added each of the chromosomal alignments separately in the supplement. The main point of Fig 5 is to illustrate the markedly reduced heterozygosity in HMS vs a well characterized CEPH pedigree.  We made minor changes to the figures that may make their intent clearer.

Reviewer 2 Report

Manuscript "A chromosome-length assembly of the Hawaiian Monk seal (Neomonarchus schauinslandi): A history of “genetic purging” and genomic stability" is very interesting.

Authors compared references to that of other HMS and to selected human samples to assess overall heterogeneity, especially in genes that may be relevant to risk of disease.
Description of material is perfect. Unfortunately, the lack in manuscript of statistical analysis.

Authors compared references. For these comparision the meta-analysis if the best tool. Lack in paper these analyses.

Paper needs major revision.

Author Response

Thank you for your comments.  We agree that to fully exploit the data we have produced for this species more analyses will be needed and are planned in the future.  The main objective of this paper is to present the genome assembly for this species and provide a glimpse at the richness of the data by highlighting a few examples.  We hope that some of the methods and approaches that we used (e.g., comparing chromosomal assemblies of related species, using gene characterization tools, producing phased variant haplotypes, and contrasting this species to human) will be useful to others planning similar studies.  

Round 2

Reviewer 1 Report

The manuscript is revised according to comments

Reviewer 2 Report

Now, all is perfect.